# Rapid Detection of Carbapenemase and Extended-Spectrum β-Lactamase Producing Gram-Negative Bacteria Directly from Positive Blood Cultures Using a Novel Protocol

**DOI:** 10.3390/antibiotics12010034

**Published:** 2022-12-26

**Authors:** Diego Fernando Josa, Ingrid Gisell Bustos, Soad Amira Yusef, Stephanie Crevoisier, Edwin Silva, Natalia López, Rafael Leal, Isabel Torres Molina, Juan Pablo Osorio, Gerson Arias, Fabián Cortés-Muñoz, Carolina Sánchez, Luis Felipe Reyes

**Affiliations:** 1Research Group Cardiovascular Medicine and Specialties of High Complexity, Fundación Clínica Shaio, Bogotá 110121, Colombia; ingridbusmo@unisabana.edu.co (I.G.B.); edwin.silva@shaio.org (E.S.); natalia.lopez@shaio.org (N.L.); rafael.leal@shaio.org (R.L.); isabel.torres@shaio.org (I.T.M.); juan.osorio@shaio.org (J.P.O.); gerson.arias@shaio.org (G.A.); fabian.cortes@shaio.org (F.C.-M.); carolina.sanchez@shaio.org (C.S.); 2Department of Critical Medicine, Fundación Clínica Shaio, Bogotá 110111, Colombia; soadyume@unisabana.edu.co (S.A.Y.); stephaniecral@unisabana.edu.co (S.C.); 3Unisabana Center of Translational Science, Universidad de la Sabana, Chía 53753, Colombia; 4Infectious Diseases Department, Fundación Clínica Shaio, Bogotá 110111, Colombia; 5Department of Critical Care, Clínica Universidad de La Sabana, Chía 250001, Colombia

**Keywords:** blood cultures, rapid diagnostic test, bloodstream infection, carbapenemases, extended-spectrum β-lactamase (ESBL), nephelometry

## Abstract

Background: Early and adequate antibiotic treatment is the cornerstone of improving clinical outcomes in patients with bloodstream infections (BSI). Delays in appropriate antimicrobial therapy have catastrophic consequences for patients with BSI. Microbiological characterization of multi-drug resistant pathogens (MDRP) allows clinicians to provide appropriate treatments. Current microbiologic techniques may take up to 96 h to identify causative pathogens and their resistant patterns. Therefore, there is an important need to develop rapid diagnostic strategies for MDRP. We tested a modified protocol to detect carbapenemase and extended-spectrum β-lactamase (ESBL) producing Gram-negative bacteria (GNB) from positive blood cultures. Methods: This is a prospective cohort study of consecutive patients with bacteremia. We developed a modified protocol using the HB&L^®^ system to detect MDRP. The operational characteristics were analyzed for each test (HB&L-ESBL/AmpC^®^ and HB&L-Carbapenemase^®^ kits). The kappa coefficient, sensitivity, specificity, positive predictive value (PPV), negative predictive value (NPV), likelihood ratios (LR) with 95% confidence intervals (CI), and reduction in identification time of this novel method were calculated. Results: Ninety-six patients with BSI were included in the study. A total of 161 positive blood cultures were analyzed. *Escherichia coli* (50%, 81/161) was the most frequently identified pathogen, followed by *Klebsiella pneumoniae* (15%, 24/161) and *Pseudomonas aeruginosa* (8%, 13/161). Thirty-three percent of isolations had usual resistance patterns. However, 34/161 (21%) of identified pathogens were producers of carbapenemases and 21/161 (13%) of extended-spectrum β-lactamases. Concordance between our HB&L^®^ modified protocol and the traditional method was 99% (159/161). Finally, identification times were significantly shorter using our HB&L^®^-modified protocol than traditional methods: median (IQR) 19 h (18, 22) vs. 61 h (60, 64), *p* < 0.001. Conclusions: Here, we provide novel evidence that using our HB&L^®^-modified protocol is an effective strategy to reduce the time to detect MDRP producers of carbapenemases or extended-spectrum β-lactamases, with an excellent concordance rate when compared to the gold standard. Further studies are needed to confirm these findings and to determine whether this method may improve clinical outcomes.

## 1. Introduction

Infections due to resistant Gram-negative bacteria (GNB) have emerged as a global public-health problem [1,2]. The impact on lives lost and costs for the health systems have been extensively studied in the medical literature [3]. Up to 40% of patients diagnosed with sepsis and septic shock due to multi-drug resistant germs die due to these infections. Additionally, survivors have prolonged hospital stays, with annual costs for the health system estimated at USD 16.7 billion in the United States alone [4,5]. Early antibiotic therapy improves the survival of patients with sepsis and bloodstream infections caused by GNB [6,7,8,9,10,11,12]. Still, the emergence of antibiotic resistance poses a growing challenge to achieving this objective. Owing to their genetic plasticity, GNBs rely on various resistance mechanisms that allow them to respond to a wide range of environmental threats; of these mechanisms, the main and most versatile is the production of extended-spectrum β-lactamases (ESBL) [13]. The global expansion of ESBLs and carbapenemases is an unprecedented event favored by the high-mobility conditions of the population [14] and the non-prudent use of antibiotics in the food and healthcare industries, among other factors [15,16].

This problem has significantly affected Latin America [17,18]. Irrespective of the type of β-lactamase involved, epidemiological surveillance studies have shown a growing prevalence of *Enterobacterials* resistance to oxyimino-cephalosporins in Latin-American hospitals, which has led to the massive use of carbapenems with the resulting emergence of isolates-resistant to these agents [19]. In Colombia, GNBs resistant to carbapenems are endemic. Colombia was the first Latin-American country to deliver isolate producers of *Klebsiella pneumoniae* carbapenemase (KPC) [20] and the first country in the world to identify isolates of KPC-producing *Pseudomonas aeruginosa* [21]. Equally, the national epidemiological surveillance has detected an endemic circulation of various carbapenemase types (i.e., OXA, NDM, and VIM) [22,23].

The use of conventional methods, such as cultures in solid media and subsequent identification through biochemical tests by automated systems, allow the differentiation of ESBL- and carbapenemase-producing Gram-negative bacilli, but the time required can be up to 24 h after obtaining the isolates in culture and up to an additional 48 h for phenotypic confirmation tests, making them very time-consuming methods. However, rapid techniques have recently been evaluated to detect multi-drug resistant GNB in blood culture samples [24]. In addition to molecular methods [25,26], biophysical techniques such as mass spectrometry, flow cytometry, laser nephelometry, immunochromatography, chemiluminescence, and bioluminescence are currently available [27]. Implementing these techniques leads to decreased hospital stays, ICU stays, costs, and mortality rates [28]. However, there are limitations, such as their high cost, the use of physical space, and the need for trained staff. More importantly, these technologies are not available in every hospital worldwide.

Unlike molecular methods, laser nephelometry (HB&L^®^ Alifax, Italy) offers excellent diagnostic performance at a lower cost. They have various uses, including sifting strategies, quantitative cultures, and antimicrobial susceptibility determination. The most significant contribution in terms of antibiotic resistance has been implementing this technology as an active surveillance strategy for the early identification of multi-drug resistant GNB carriers in rectal swabs [29,30,31]. Its use in blood samples has been limited [32,33]. We hypothesize that this technique is efficient for rapidly detecting multi-drug-resistant GNB in patients with bacteremia. Therefore, the objective of this study is to evaluate the operational characteristics resulting from a new modified protocol designed to detect Gram-negative bacteria producing carbapenemases and extended-spectrum β-lactamases from positive blood cultures using HB&L^®^ laser nephelometry, compared with the reference standard.

## 2. Materials and Methods

This is a prospective cohort observational study of diagnostic tests in which the operational characteristics are evaluated, in addition to the degree of concordance and the diagnostic time resulting from the combination of the Bact/Alert 3D^®^, HB&L^®^ laser nephelometry, and Vitek2^®^ techniques compared with the reference standard in the development of a new protocol for the direct assembly of positive blood cultures in a high-complexity hospital in the city of Bogota, Colombia. The research protocol was approved by the Shaio Clinic Foundation’s Ethics and Research Committee (Memorandum of Approval No. 273). Obtaining informed consent was unnecessary because of the absence of direct intervention in patients and the observational characteristics of the study.

The HB&L^®^ equipment (Alifax, Polverara, Italy) is designed to perform quantitative kinetic counting in colony-forming units per milliliter (CFU/mL) in urine samples, biological fluids, and rectal swabs by laser nephelometry methodology. For the detection of carbapenemases and ESBL in rectal swabs, it has kits which contain a reagent composed of a mixture of antibiotics, including vancomycin; an antifungal; and carbapenems to eliminate the accompanying microbiota in the rectal samples, and allow the growth of GNB resistant to carbapenems. In our study, under our design, we apply this kit to detect carbapenemases and ESBL in positive-blood-culture bottles and evaluate their performance.

### 2.1. Sample Processing

All blood cultures sent to the microbiology laboratory from 1 July 2017 to 31 March 2019 were collected consecutively and prospectively. All positive blood cultures with GNB identified via direct Gram-stain microscopy were included. The only exclusion criterion in our study was blood cultures in which the presence of Gram-positive cocci, Gram-positive bacilli, and yeast was observed.

#### 2.1.1. Setting up Blood Cultures for Laser Nephelometry Using HB&L-ESBL/AmpC^®^ and HB&L-Carbapenemase^®^ Kits (New Protocol)

With the help of a syringe, a blood sample was taken from the positive-blood-culture bottle, and two drops of blood were released into a plastic tube with 2 mL of 0.9% saline solution and mixed using a vortex. Based on this prepared suspension, 200 μL was released into the green-lidded vial from the HB&L-ESBL/AmpC^®^ kit, together with 200 μL of the reagent from the kit and another 200 μL into the red-lidded vial from the HB&L^®^ Carbapenemase kit with 200 μL of the specific reagent (including vancomycin, an antifungal, and carbapenems); the vials were then deposited in the automated HB&L^®^ system in the pre-established programs ESBL and carbapenemase, respectively. The results were subsequently read after 6 h (Figure 1).

#### 2.1.2. Setting up the Tests through the Conventional Method

Culturing was performed through the conventional method in solid culture mediums. For this, a positive blood-culture sample was extracted, releasing a drop into blood agar and a drop into MacConkey agar. Subsequently, culturing was performed by dropping using a handle, and cultures were incubated at 37 °C for 24 h. The following day, colonies were subjected to biochemical identification and susceptibility testing using the Vitek2^®^ automated system (Biomerieux^®^) with GNB and AST-272 cards.

#### 2.1.3. Phenotypic Tests for Confirmation of ESBL, AmpC, and Carbapenemase-Producing GNB

Isolates with antimicrobial susceptibility test with MICs greater than or equal to 1 ug/mL for ertapenem or greater than or equal to 2 ug/mL for imipenem or meropenem (according to CLSI 2020) [34], underwent confirmatory tests for the detection of carbapenemase production, such as the Hodge test, synergy tests with EDTA disks, and synergy tests with boronic acid disks.

For the confirmation of ESBL and AmpC, the double-disc test with third-generation cephalosporin discs combined with clavulanic acid was used as the reference method, according to CLSI standards [34]. In the study period of this investigation, the Hodge test was the method used to confirm the presence or absence of carbapenemases, and synergy tests with EDTA discs and boronic-acid discs were used to differentiate the type of carbapenemase present (serine or metallobetalactamase).

#### 2.1.4. Test Controls

The following control strains were used in setting up each test: *K. pneumoniae* ATCC BAA 1705 (*bla*KPC+), *K. pneumoniae* ATCC BAA 2146 (*bla*NDM+), ESBL-producing *K. pneumoniae* ATCC 700603, *Escherichia coli* ATCC 25922, and *P. aeruginosa* ATCC 27853.

### 2.2. Statistical Analysis

The operational characteristics were analyzed separately for each test (HB&L-ESBL/AmpC^®^ and HB&L-Carbapenemase^®^ kits, Alifax, Padua, Italy). Kappa coefficient, sensitivity, specificity, positive predictive value (PPV), negative predictive value (NPV), and positive and negative likelihood ratios (LR) with 95% confidence intervals (CI) were calculated. In addition, the positivity time (identification of multidrug-resistant BGN) was evaluated using the proposed identification protocol and the diagnosis time used by the conventional method. Data were analyzed using SPSS 28, IBM, USA.

## 3. Results

We included 161 positive-blood-culture bottles with GNB from 96 subjects hospitalized owing to different diagnoses (Table 1).

The most commonly identified germ was *E. coli* (81/161, 50.1%), followed by *K. pneumoniae* (24/161, 14.9%) (Figure 2). A certain degree of antimicrobial resistance was observed in 67% (109/161) of the samples, with carbapenems resistance being the most frequently identified (34/161, 21%), followed by ESBL (21/161, 13%) and amplified-spectrum β-lactamase (18/161, 11%). It is essential to highlight that a common sensitivity profile was observed in 33% (53/161) of the samples (Table 2).

### 3.1. Operational Characteristics of the HB&L-ESBL/AmpC^®^ Test

Sensitivity of the HB&L-ESBL/AmpC^®^ test in blood cultures was 95.4% with 100% specificity. Predictive values VPP and VPN were 100% and 98.3%, respectively (Table 3). Compared with the conventional method, concordance, assessed through the kappa correlation coefficient, was 0.92 with a 97.5% concordance rate.

### 3.2. Operational Characteristics of the HB&L Carbapenemase^®^ Test

All operational characteristics (S, E, VPN, and VPP) from the vial for carbapenemase were 100% (Table 3), with a correlation coefficient of 100%.

### 3.3. Time to the Identification of Multi-Drug Resistant Germs

The median (IQR) time to identify GNB in the Bact/Alert 3D^®^ system was 13 h (12, 16). When the modified HB&L^®^ method for positive blood-culture samples was compared with the conventional identification method, the presumptive positivity for the identification of carbapenemase-producing GNB was much faster when the modified HB&L was used (19 h [18, 22] vs. 61 h [60, 64]; *p* < 0.001), including the positivity time from the Bact/Alert 3D^®^ system. In other words, a 42-h reduction was achieved in the early recognition of carbapenemase and/or oxyimino-cephalosporin resistance (Figure 3). At the time of identifying carbapenemase-producing GNB, their presence was confirmed through the phenotypic testing Hodge test and differentiation of the type of enzyme through synergy tests with EDTA discs and boronic-acid discs (data not included).

## 4. Discussion

Our new protocol for the rapid detection of ESBL and carbapenemases demonstrated high sensitivity, which is why it can be used in blood-culture samples with excellent results. A 42-h reduction was noted in identifying carbapenem and oxyimino-cephalosporin resistance in GNB from blood cultures (Figure 3). The reduction in time positively impacts two main objectives: (1) initiation of targeted antibiotic therapy according to the type of resistance identified and (2) reduction in direct hospitalization costs by having these results in less time. Additionally, it was found that compared with the conventional method, the combination of the Bact/Alert 3D^®^, HB&L^®^, and Vitek2^®^ techniques resulted in a concordance of 100% for the early detection of carbapenemase and a concordance of 95% in the detection of phenotypic resistance to oxyimino-cephalosporins (Table 3). This time saved can positively impact the clinical outcomes of such patients.

The early initiation of effective antibiotic therapy predicts the outcome of bacteremia due to GNB [6,8,9,10,11,12]. Tumbarello et al. [8] showed that the initiation of inappropriate antibiotic treatments was a strong predictor of mortality in patients with bacteremia due to ESBL-producing GNB (59.5% vs. 18.5%; OR: 2.28; 95% CI: 1.76–3.22; *p* < 0.001). A meta-analysis conducted by Kohler et al. reported a mortality of 46% in patients with infections caused by carbapenemase-producing GNB, which was even higher in those receiving inappropriate treatment (10% higher, OR: 1.28; 95% CI: 1.04–1.58; *p* = 0.02) [12]. Kang et al. reported an 11% reduction in the total crude mortality when adequate empiric antibiotic treatment was administered [6]. Likewise, the INCREMENT cohort reported a 22% increase in mortality resulting from the initiation of inappropriate empiric therapy [11]. Therefore, it is evident that tests are necessary to detect multi-drug-resistant germs to ensure adequate antibiotic treatment rapidly.

In various studies, rapid detection tests showed a reduction in hospital and ICU stays, mortality rate, and costs [25,26,27]. Perez et al. incorporated the rapid identification of pathogens through matrix-assisted laser desorption/ionization time (MALDI-TOF) and susceptibility through BD Phoenix^TM^, with a decrease by more than 50% in the microbiological identification times (40.6 vs. 14.5 h; *p* < 0.001), adjustment to effective treatment (89.7 vs. 32 h; *p* < 0.001), reduced hospital stay (23.3 vs. 15.3 days; *p* = 0.001) and ICU stay (16 vs. 10.7 days; *p* = 0.008), and 30-day mortality (21% vs. 8.9%; *p* = 0.01) [28]. Sakarikou et al. also used MALDI-TOF for identification purposes as well as VITEK-2^®^ for susceptibility tests in blood-culture samples with GNB. In this case, the sample was taken directly from the blood culture without moving it to a solid medium, saving 8 h compared with the conventional method (5 vs. 11 h; *p* ≤ 0.001, without taking time for blood-culture positivity into account) and with a concordance of 98.5% [35]. This evidence confirms that identifying multi-drug resistant bacteria early makes it possible to provide a rapid, targeted therapy and improve clinical outcomes. Unfortunately, the MALDI-TOF availability is limited to a few hospitals owing to the high initial installation cost.

PCR-based techniques conducted microbiological identification in 1–2 h. However, antibiotic susceptibility is limited to the genes included in each panel, such as the Film Array^®^ platform (Biofire^®^, Salt Lake City, UT, USA), where the BCID 1 or 2 panel is still used in some Latin-American countries, detects the presence of different carbapenemase genes. Additionally, in some of these techniques, a decreased efficiency is observed in polymicrobial infections, thereby making the combination of additional techniques necessary [26,27,28]. On the other hand, the Bact/Alert 3D^®^, VITEK^®^, and HB&L^®^ techniques are less costly and do not require specialized training, resulting in easy implementation in hospitals. Studies such as those conducted by Hogan and Höring [36,37] on GNB-positive blood culture involved direct inoculation in evaluating susceptibility through VITEK-2^®^; the concordance with the conventional method was over 95% in both studies. However, false susceptibilities to carbapenemases were documented, constituting a major error. Recently, Athamna et al. used Uro4 HB&LTM laser nephelometry, compared with VITEK-2^®^, in their study on ESBL/AmpC susceptibility in Enterobacterials, with 91.3% concordance. However, carbapenemase identification was not included, and for *P. mirabilis*, the concordance was only 58.3% with zero sensitivity [32]. Other studies were performed by rapid incubation, such as the EUCAST RAST test, which uses discs of different antibiotics in short incubation times (4 to 6 h) [38].

Currently, there are new Rapid ESBL NP^®^ colorimetric tests (Liofilchem, Italy) based on the hydrolysis of cefotaxime which detect the presence of ESBL very quickly, but to perform them directly from positive blood cultures requires additional kits which allow obtaining the bacterial sediment but increase the cost of testing and further steps in assembly [39]. Additionally, the immunochromatography tests for ESBL and carbapenemases, CARBA 5 and NG-Test^®^ CTX-M (NG-Test^®^, Guipry, France) are performed from colonies of the isolates in solid culture [40] and have a higher cost than the laser nephelometry tests (USD 14 versus USD 6 respectively).

In contrast, our study integrated the Bact/Alert 3D^®^, VITEK^®^, and HB&L^®^ techniques in GNB-positive blood cultures, with direct inoculation using both kits (ESBL/AmpC^®^ and Carbapenemase^®^) (Figure 1). Identification time and susceptibility decreased by 42 h compared with the conventional methodology, which involves a significant decrease in the identification times which could potentially improve the clinical outcomes of patients with bacteremia due to GNB. Occasionally, the efficiency was 100% in all operational characteristics using the HB&L Carbapenemase^®^ kit (Table 2), including the identification of serine carbapenemases and Metallo-β-lactamases. Thus, this study opens the way for new low-cost strategies aimed at rapidly detecting these pathogens; however, it must be assessed whether the routine use of this protocol impacts the clinical outcomes of patients with bacteremia.

The strengths and weaknesses of our study are noteworthy. Despite being a unicentric study, the incidence of antibiotic resistance resembles that at the international level; thus, similar endemic populations with multi-drug resistance could benefit from this methodology and the results presented in this manuscript. Two other important limitations are the small number of samples included in the study and the lack of genomic sequencing or PCR for the molecular identification of ESBL, AmpC, and carbapenemases. Concerning cost, the systematic use of HB&L^®^ for all GNB-positive blood cultures could initially have a negative impact; however, it could be considered cheap when comparing it with alternative quick tests and evaluating the impact of the optimal initiation of therapy in terms of other cost-effective scenarios such as hospital stay, ICU stay, antibiotic savings, and mortality rate. However, the assessment of this test’s economic and clinical impacts was out of this study’s scope and needs further study in other research. The phenotypic diagnosis led to limitations owing to the presence of combinations in increasing resistance mechanisms; however, in these limited-resource scenarios, we believe this methodology is one of the best diagnostic options. Moreover, it provides another fast diagnostic opportunity using concentrated biomass in the HB&L^®^ equipment to apply other methods, such as immunochromatography or lateral flow for detecting carbapenemases and ESBL, as demonstrated in recent studies of direct protocols with good sensitivity and specificity [40].

## 5. Conclusions

The direct assembly of our protocol from bottles of positive blood cultures demonstrated high sensitivity and specificity in the early detection of ESBL and carbapenemase compared to conventional methods. The implementation of our new protocol reduces the reporting time of these multi-resistant microorganisms by up to 42 h compared to the traditional method, which allows decisions to be made in targeted treatment therapies. It is also an effective strategy to control the spread of this type of multi-resistant microorganisms in the hospital environment, since it allows effective isolation and cohortization measures to be taken.

This new protocol has a lower cost than molecular tests and immunochromatographic tests, with equal speed, and can be implemented in different types of hospital institutions of low and medium complexity, where currently only traditional methods are used. Additional studies are required to assess this test’s economic and clinical impacts.

## Figures and Tables

**Figure 1 antibiotics-12-00034-f001:**
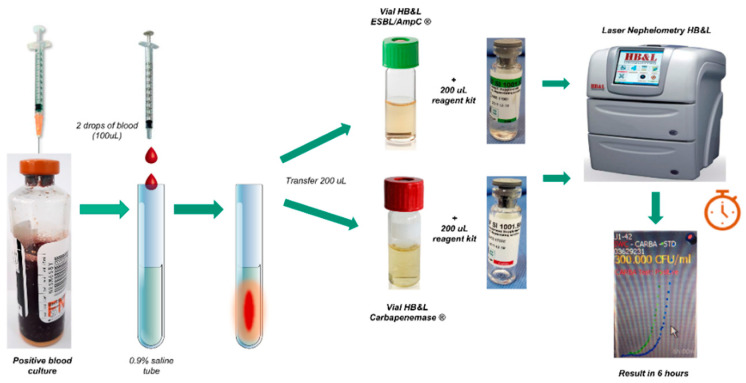
Processing of positive-blood-culture samples by ESBL/AMPC and HB&L Carbapenemase kit tests. Green-cap vial: vial for detecting Gram-negative bacilli with ESBL extended-spectrum beta-lactamases. Red-cap vial: vial for detecting Gram-negative bacilli carriers of carbapenemases.

**Figure 2 antibiotics-12-00034-f002:**
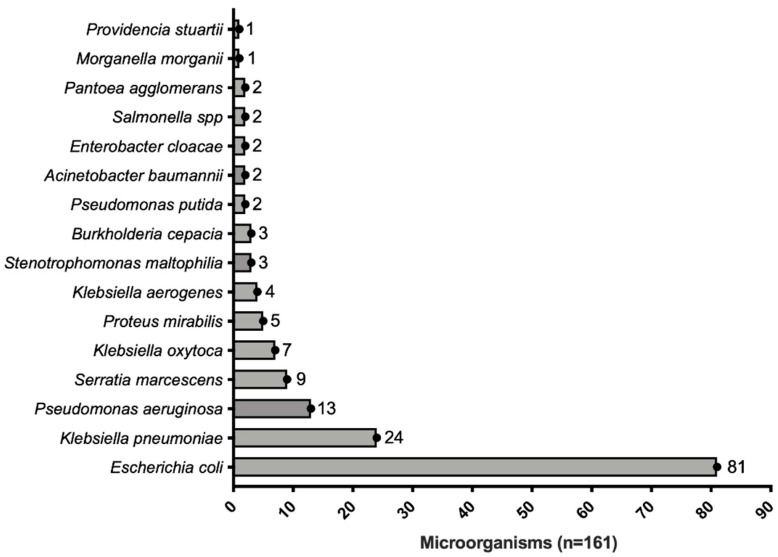
Distribution of microorganisms retrieved from patients with bacteremia.

**Figure 3 antibiotics-12-00034-f003:**
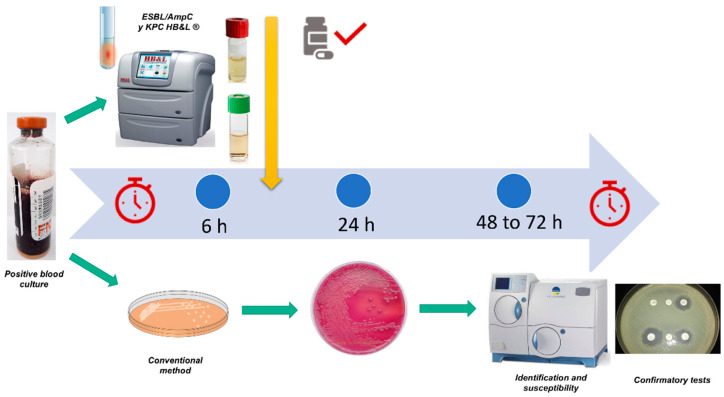
Comparison of response times for each technique (turnaround time). Above the timeline: the tests of our study applied to blood-culture results were obtained around 6 h. Below the timeline: the time required by the conventional method is denoted, with MacConkey agar seeding, then after 24 h susceptibility tests on Vitek 2, and at 48 to 72 h, the result of carbapenemase confirmation tests (Hodge test, synergy tests with EDTA discs and boronic-acid discs.

**Table 1 antibiotics-12-00034-t001:** General characteristics of patients with bacteremia.

Characteristic	Total (*n* = 96)
Sex, *n* (%)	
Men	43 (44.8)
Women	53 (55.2)
Age, median (IQR)	73 (59, 80)
Comorbidities, *n* (%)	
Arterial hypertension	53 (55.2)
Coronary disease	16 (16.7)
COPD	18 (18.8)
OSAHS	8 (8.3)
Diabetes Mellitus	33 (33.4)
Obesity	12 (12.5)
Chronic kidney disease	28 (29.2)
Severity, median (IQR)	
APACHE II	12.5 (8, 20.75)
SOFA	4 (2.25, 6.75)
Origin of samples, *n* (%)	
Hospitalization	48 (45.8)
Intensive care unit	40 (41.7)
Emergencies	11 (11.5)
Source of infection, *n* (%)	
Community-acquired	61 (63.5)
Nosocomial infection	33 (34.4)
Bacteremia, *n* (%)	
Primary	7 (7.3)
Secondary	89 (92.7)
Origin, *n* (%)	
Urine	45 (46.9)
Abdomen	12 (12.5)
Bile duct	10 (10.4)
Lung	8 (8.3)
Time, hours (median [IQR])	
Bact/Alert 3D^®^ positivity	13 (12, 16)
HB&L^®^ positivity	19 (18, 22)
Conventional culture^®^	61 (60, 64)

APACHE II: Acute Physiology and Chronic Health Evaluations II; COPD: Chronic Obstructive Pulmonary Disease; OSAHS: Obstructive Sleep Apnoea-Hypopnoea Syndrome; SOFA: Sequential Organ Failure Assessment, IQR: Interquartile Range.

**Table 2 antibiotics-12-00034-t002:** Distribution of resistance profiles and concordance between the conventional identification method and antibiogram and the method proposed by HB&L.

	Microorganism	(*n*)	Conventional Method/HB&L Method						
			Sensitive	IRT	ASBL	ESBL	Repressed AmpC	Unrepressed AmpC	Resistance to Carbapenems
**Enterobacterials**	*E. coli*	81	34/34	13/13	15/15	17/17			2/2
	*K. pneumoniae*	24	6/6		1/1	2/2			15/15
	*S. marcescens*	9					4/4	5/5	
	*K. oxytoca*	7	4/4		2/2	1/1			
	*P. mirabilis*	5	5/5						
	*K. aerogenes*	4					1/1	1/1	2/2
	*E. cloacae*	2						2/2	
	*Salmonella* spp.	2	1/1			1/1			
	*P. agglomerans*	2							2/2
	*M. morganii*	1						1/1	
	*P. stuartii*	1					1/1		
**NF GNB**	*P. putida*	2							2/2
	*P. aeruginosa*	13					5/5	3/1	5/5
	*B. cepacia*	3	2/2						1/1
	*S. maltophilia*	3							3/3
	*A. baumannii*	2							2/2

Comparative data between the two methods, according to the comparison of antibiogram data obtained by the Vitek 2 automated system. IRT: Inhibitor Resistant TEM; ASBL: extended spectrum beta-lactamase; ESBL; extended-spectrum beta-lactamase; NF GNB: non-fermenting Gram-negative bacilli.

**Table 3 antibiotics-12-00034-t003:** Operational characteristics of the ESBL/AmpC and HB&L Carbapenemase kit tests.

	ESBL/AmpC^®^ Vial				Carbapenemase^®^ Vial			
	Positive	Negative	Total		Positive	Negative	Total	
Positive conventional culture	61	2	63	S (95%)	34	0	34	S (100%)
Negative conventional culture	0	98	98	E (100%)	0	127	0	E (100%)
Total	61	100	161		34	127	161	
	VPP 100%	VPN 98%			VPP 100%	VPN 100%		

S: sensitivity; E: specificity; VPP: positive predictive value; VPN: negative predictive value.

## Data Availability

After contacting the corresponding author.

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
