# Peer review of "Rapid Detection of Carbapenemase and Extended-Spectrum β-Lactamase Producing Gram-Negative Bacteria Directly from Positive Blood Cultures Using a Novel Protocol"

_antibiotics, 2022, doi:10.3390/antibiotics12010034_

Round 1

Reviewer 1 Report

Although authors have tried to improve their method and have claimed that testing time is reduced for the detection Carbapenemase and Extended-Spectrum β- 2 lactamase.  However, there are a few observations.

1.       Abstract must have more technical but specific details. The use of references in the abstract is not recommended. The abstract is lacking to understand of how has this protocol improved. What is modified and what process has been improved.

2.       Introduction needs improvement. The novelty must be justified. What is required to improve the method? What were the deficiencies of the current method which is highly demanded to be improved?

3.       Statistical analysis is not written in a standard way in the methodology section. More in-depth cost and technical analysis comparison is required to make the decision about the protocol.

4.       The error between the methods is not calculated. The sensitivity analysis is missing.

5.       The design of the study is poor and lacks depth.

6.       Figure 1, 3, and Table 2 need to be clear and more informative with more technical details. In table two do the operational characteristics of two vials make any difference? It seems both have 90-100% results. It doesn’t make big difference.

7.       Discussion section should ref to figures and tables to justify their outcomes with possible explanations.

8.       Better to highlight through a diagram what has been improved in the method and justify that with data and good statistical analysis. Sensitivity analysis is very important.

9.       Conclusions must be written in one paragraph to conclude clearly what is significant outcome has been made through this study. What was the limitation of this work and what are the future recommendations?

10.   The number of replicates and the standard errors is also missing. 

Author Response

  1. The abstract should have more technical but specific details.

R/ More technical details were added.

The use of references in the abstract is not recommended.

R/There are no references in the abstract.

  1. The introduction needs improvement. The novelty must be justified. What is required to improve the method? What were the deficiencies of the current method that is highly demanded to be improved?

R/ Improvements were made in the introduction, which specified the current conventional method's present deficiency and why we want to improve with our new protocol.

  1. Statistical analysis is not written in a standard way in the methodology section.

R / Following your recommendation, we have edited the text.

A more in-depth cost and technical analysis comparison is required to decide on the protocol.

R/ We thank the reviewer for this critical comment. However, the cost analysis was not one of the objectives of our study. Thus, we propose evaluating or addressing this in future research. However, we have included some additional wording in the discussion.

"With respect to cost, the systematic use of HB&L® for all GNB-positive blood cultures could initially have a negative impact; however, it could be considered cheap when comparing it with alternative quick tests and evaluating the impact of the optimal initiation of therapy in terms of other cost-effective scenarios such as hospital stay, ICU stay, antibiotic savings, and mortality rate. However, the assessment of this test's economic and clinical impacts was out of the scope of this study and needed to be further studied in other research."

  1. The error between the methods is not calculated. Missing sensitivity analysis.

R/ The sensitivity analysis that was carried out is included in the results section.

  1. The study design is poor and lacks depth.

R/ We appreciate your comment about the study design. We have included text to highlight our study design in the manuscript.

  1. Figure 1, 3 and Table 2 should be clear and more informative with more technical details. In table two, do the operating characteristics of two vials make a difference? It seems that both have 90-100% results. It doesn't make a big difference.

R/ Corrections and modifications were made in the figures and table 2.

Indeed, Table 2 shows the operational characteristics of two vials. As explained in methods, one vial is for detecting ESBL, and the other vial is for detecting BGN resistant to carbapenems, both of which achieve excellent detection results. The text has been updated.

  1. The discussion section should refer to figures and tables to justify your results with possible explanations.

R/ Figures and tables are referenced.

  1. Better to highlight through a diagram what has been improved in the method and justify it with data and a good statistical analysis. Sensitivity analysis is very important.

R/ The analysis of the sensitivity obtained from our new protocol is added in several sections of the manuscript following your recommendation. The diagram that demonstrates what we improve with our method is already added in the text, which offers much faster results than the conventional method, significantly impacting time. (see figure 3)

  1. The conclusions should be written in one paragraph to clearly conclude what is the significant result that has been achieved through this study. What was the limitation of this work and what are the future recommendations?

R/ The conclusions have been modified following your recommendation.

  1. The number of repetitions and the standard errors are also missing.

R/ There were no repetitions carried out in the study.

Reviewer 2 Report

The topic of the paper is of interest but a major revision is necessary.

Major comments:

1. Reference methods for ESBL and AmpC detection are not described in material and methods section.

2. Clinical variables are not necessary for this type of study. Therefore, they should be removed (lines 149-153). Lines 168-178 Clinical variables have no meaning in this type of study. These lines should be removed.

3. Table 2: Please provide list of abbreviations and table footnote. Methods for differentiation of β-lactamase production are not described in the study. How do you differentiate ESBL, repressed/unrepressed AmpC? Resistance to carbapenemase is incorrect. Do you mean resistance to carbapenems or carbapenemase production? Please correct here and elsewhere. For carbapenemases, differenziation of serine-carbapenamase and metallo-β-lactamases should be reported.

4. The first part of discussion regarding importance of rapid diagnostic is already described in introduction section. This part should be shortened. Conversely, the discussion section should be enriched with a comparison of the performance of the assays here described with those recently described in the literature. What are the advantages (performance, ease of execution and cost) compared to already evaluated methods? (e.g. colorimetric methods doi: 10.1007/s10096-021-04385-1, doi: 10.1128/JCM.00709-19; immunochromatographic methods doi: 10.1093/jac/dkac230; molecular methods doi: 10.3390/antibiotics11020138).

Minor comments

1. LINE 74 “time required can be 80 h after obtaining the isolate”. At most 24 h, please clarify.

2. LINE 120 Extraction of what? Please clarify.

3. line 137-138. Colonies were subjected to biochemical identification and susceptibility testing

4. Lines 140-144 Hodge test is a recommended method for carbapenemase detection. Please specify according to which guidelines.

5. Lines 168-178 Clinical variables have no meaning in this type of study. These lines should be removed.

6. Lines 181 antimicrobial

7. Line 182. carbapenems resistance or carbapenemase production

8. Lines 183-184 amplified-spectrum β-lactamase. Do you mean AmpC?

9. Lines 282 Molecular identification of ESBL and AmpC is a further limitation.

10. Line 295 Add a reference (see doi: 10.1093/jac/dkac230)

Author Response

Reviewer 2

Important comments:

  1. Reference methods for the detection of ESBL and AmpC are not described in the materials and methods section.

R/ A paragraph has been added to describe the method to confirm ESBL and AmpC with the double-disc test with third-generation cephalosporin discs combined with clavulanic acid, according to CLSI standards.

  1. Clinical variables are not necessary for this type of study. Therefore, they should be removed (lines 149-153). Lines 168-178 Clinical variables have no meaning in this type of study. These lines should be removed.

R/Clinical variables were eliminated following your recommendation.

  1. Table 2: Provide a list of abbreviations and a footnote to the table.

R/ Following your recommendation, we have added a section describing abbreviations in the table.

Methods for the differentiation of β-lactamase production are not described in the study. How do you differentiate ESBL, repressed/unrepressed AmpC?

R/ To differentiate ESBL and AmpC, the double disc test with clavulanic acid was used. And AmpC repressed and not repressed; we only determine it according to the profile given by the antibiogram by an automated system, in which, if there is evidence of resistance (not repressed) or no resistance (repressed) to third generation cephalosporins.

Carbapenemase resistance is incorrect. Do you mean carbapenem resistance or carbapenemase production? Please correct here and elsewhere.

R/ We have edited the text accordingly.

For carbapenemases, the differentiation of serine-carbapenamase and metallo-β-lactamases should be reported.

R/ The method used in this study only identified the presence or absence of carbapenemase and β-lactamases. Thus, it does not allow differentiating between serine-carbapenamase and metallo-β-lactamases

  1. The first part of the discussion on the importance of rapid diagnosis is already described in the introduction section. This part should be shortened.

 Rather, the discussion section should be enriched with a comparison of the performance of the trials described here with those recently described in the literature. What are the advantages (performance, ease of execution and cost) compared to the methods already evaluated? (for example, colorimetric methods doi: 10.1007/s10096-021-04385-1, doi: 10.1128/JCM.00709-19; immunochromatographic methods doi: 10.1093/jac/dkac230; molecular methods doi: 10.3390/antibiotics11020138).

R/ The first paragraph of the discussion has been edited based on your recommendation.

Minor comments:

  1. LINE 74 "the necessary time can be 80 h from the isolate obtaining". At most 24 hours, please clarify.

R/ We have clarified that the time needed for this protocol is 48 hours.

  1. LINE 120 Extraction of what? Please clarify.

R/ The text has been edited.

  1. line 137-138. Colonies were subjected to biochemical identification and susceptibility tests.

R/ The text has been edited.

  1. Lines 140-144 The Hodge test is a recommended method for detecting carbapenemases. Please specify under which guidelines.

R / Although it is true that the latest CLSI guidelines no longer recommend the Hodge test, we want to clarify that it was still valid during the study period of this research. The text has been edited.

  1. Lines 168-178 Clinical variables have no meaning in this type of study. These lines should be removed.

R/ The text has been edited.

  1. Lines 181 antimicrobial

R/ The text has been edited.

  1. Line 182. Resistance to carbapenems or production of carbapenemases.

R/ The text has been edited.

  1. Amplified spectrum β-lactamase lines 183-184. Do you mean AmpC?

R/ Extended-spectrum beta-lactamase is another of the categories of beta-lactam resistance profiles, which only includes resistance to first-generation and second-generation cephalosporins. This profile is widespread in isolates in Latin America, which is why it is a commonly used term.

  1. Lines 282 Molecular identification of ESBL and AmpC is another limitation.

R/ The text has been edited.

  1. Line 295 Add a reference (see doi: 10.1093/jac/dkac230)

R/ The reference has been added.

Reviewer 3 Report

The Authors accurately describe the usefulness of a modified protocol for HB&L detection of AMR (carbapenemase and ESBL). Overall, the study is interesting and well conducetd. Some minor check for english language is needed.

Author Response

Reviewer 3

The authors accurately describe the utility of a modified protocol for ADR detection by HB&L (carbapenemase and ESBL). Overall, the study is interesting and well done. A minor English language verification is needed.

R/ Thank you for your comment about our study. A native English speaker has edited the language.

Round 2

Reviewer 1 Report

Improve conclusion carefully.

Author Response

The conclusions have been revised and improved following your recommendations.

Reviewer 2 Report

The manuscript has been improved however the following changes should be made.

Paragraph 2.1.3. is still unclear and should be improved:
carbapenemases detection should be conducted not only on carbapenems-resistant strains (according to EUCAST cut off meropenem and ertapenem MICs >0.125 mg/L). Refer to CLSI or EUCAST recommendations for carbapenemases screening. (please add reference)
Testing for carbapenemases is unclear: if synergy testing with inhibitors was not used this method should be removed. Only the modified Hodge test should be reported.

Author Response

We thank the reviewer for this careful revision and comment. Following your recommendation, we have included this clarification in paragraph 2.1.3, and the CLSI reference has been added.

Moreover, we have clarified in the text that the EDTA test and boronic inhibitors were used along with the Hodge test to differentiate the type of carbapenemase present (serine or metallobetalactamase). However, the study does not find quantification data for each type of enzyme.